# The Profile of Circulating Blood microRNAs in Outpatients with Vulnerable and Stable Atherosclerotic Plaques: Associations with Cardiovascular Risks

**DOI:** 10.3390/ncrna8040047

**Published:** 2022-06-29

**Authors:** Andrey N. Rozhkov, Dmitry Yu. Shchekochikhin, Yaroslav I. Ashikhmin, Yulia O. Mitina, Veronika V. Evgrafova, Andrey V. Zhelankin, Daria G. Gognieva, Anna S. Akselrod, Philippe Yu. Kopylov

**Affiliations:** 1World-Class Research Center “Digital Biodesign and Personalized Healthcare”, I. M. Sechenov First Moscow State Medical University, 119991 Moscow, Russia; dashkagog@mail.ru (D.G.G.); fjk@inbox.ru (P.Y.K.); 2Department of Cardiology, Functional and Ultrasound Diagnostics, N.V. Sklifosovsky Institute of Clinical Medicine, I. M. Sechenov First Moscow State Medical University, 119991 Moscow, Russia; agishm@list.ru (D.Y.S.); veronikaevgrafova@yandex.ru (V.V.E.); 7402898@mail.ru (A.S.A.); 3International Medical Cluster, 40 Bolshoy Boulevard Skolkovo Innovation Center, 121205 Moscow, Russia; ya.ashikhmin@gmail.com; 4Skolkovo Institute of Science and Technology, 121205 Moscow, Russia; yulia.mitina@skolkovotech.ru; 5Federal Research and Clinical Center of Physical-Chemical Medicine of Federal Medical Biological Agency, 119435 Moscow, Russia; zhelankin@rcpcm.org

**Keywords:** microRNA, atherosclerosis, vulnerable plaque, cardiovascular risk, coronary computed tomography angiography, coronary calcinosis, miR-143-3p, miR-181b-5p, miR-126-5p, miR-150-5p

## Abstract

Non-coding RNAs reflect many biological processes in the human body, including athero-sclerosis. In a cardiology outpatient department cohort (N = 83), we aimed to compare the levels of circulating microRNAs in groups with vulnerable plaques (N = 22), stable plaques (N = 23) and plaque-free (N = 17) depending on coronary computed tomography angiography and to evaluate associations of microRNA levels with calculated cardiovascular risks (CVR), based on the SCORE2 (+OP), ACC/AHA, ATP-III and MESA scales. Coronary computed tomography was performed on a 640-slice computed tomography scanner. Relative plasma levels of microRNA were assessed via a real-time polymerase chain reaction. We found significant differences in miR-143-3p levels (*p* = 0.0046 in plaque-free vs. vulnerable plaque groups) and miR-181b-5p (*p* = 0.0179 in stable vs. vulnerable plaques groups). Analysis of microRNA associations with CVR did not show significant differences for SCORE2 (+OP) and ATPIII scales. MiR-126-5p and miR-150-5p levels were significantly higher (*p* < 0.05) in patients with ACC/AHA risk >10% and miR-145-5p had linear relationships with ACC/AHA score (adjusted *p* = 0.0164). The relative plasma level of miR-195 was higher (*p* < 0.05) in patients with MESA risk > 7.5% and higher (*p* < 0.05) in patients with zero coronary calcium index (*p* = 0.036). A linear relationship with coronary calcium was observed for miR-126-3p (adjusted *p* = 0.0484). A positive correlation with high coronary calcium levels (> 100 Agatson units) was found for miR-181-5p (*p* = 0.036). Analyzing the biological pathways of these microRNAs, we suggest that miR-143-3p and miR-181-5p can be potential markers of the atherosclerosis process. Other miRNAs (miR-126-3p, 126-5p, 145-5p, 150-5p, 195-5p) can be considered as potential cardiovascular risk modifiers, but it is necessary to validate our results in a large prospective trial.

## 1. Introduction

Cardiovascular diseases (CVD), of which coronary artery disease (CAD) and stroke are the most common, are the leading cause of death in the United States and Europe and pose a huge socioeconomic burden on patients and healthcare systems [1,2].

As an initial step in the diagnosis of CVD, risk factors profile and functional tests are often evaluated [3]. The role of population-based methods of risk stratification (SCORE, Framingham, ACC/AHA scales and others) is limited in the field of personalized medicine, especially among young or elderly patients. Nowadays, methods of additional risk stratification frequently get into routine clinical practice [4].

In preventive cardiology, the main directions for the development of additional cardiovascular risks (CVR) stratification tools are the heart and blood vessels imaging technologies, as well as a plethora of biomarkers. X-ray diagnostic methods include the quantitative assessment of coronary calcium and coronary computed tomography angiography (CTA), which today are in focus in modern clinical guidelines [3,4,5].

In recent years, the question of the participation of microRNAs (miRNAs) in various biological processes has been actively studied. MiRNAs are a class of short non-coding RNAs that regulate gene expression at the post-transcriptional level. Many studies focused on the important role of miRNAs in the initiation and progression of CVD [6]. It has been shown that they are involved in the regulation of local immune processes in atherosclerosis and promoting or preventing vascular remodeling [7,8]. Changes in the levels of certain microRNAs are associated with major adverse cardiovascular events in patients with CAD [9,10,11].

Despite the growing interest in the study of circulating miRNAs in cardiology, new biomarkers are still not included in cardiovascular risk assessment systems, which does not allow making clinically significant decisions based on their assessment. Most of the miRNA studies of cardiovascular disease pathogenesis were fundamental; however, it is worth noting that several works have been carried out to date, focused on determining the risk of developing cardiovascular complications (CVC) based on an integral assessment of demographic and clinical signs and novel miRNAs. The first prospective studies assessed the associations of miRNA levels with cardiovascular risk and myocardial infarction incidence or mortality. In particular, miR-126, miR-223 and miR-197 were associated with the risk of infarction, and miR-133 and a group of 5 miRNAs were associated with the risk of death [12,13]. From a clinical point of view, additional interest is the possibility of assessing the vulnerability of an atherosclerotic plaque since it allows a more accurate assessment of the risks of vascular events and optimization of therapeutic strategy [14,15,16]. Several studies have shown the possible role of miR-122-5p, miR-223-3p, miR155-5p, miR-483-5p and miR-451a in the process of plaque damage; however, they were evaluated during the invasive intervention [17,18]. Other studies have focused on the non-invasive assessment of associations of CTA data with cardiovascular risk; however, they have not considered the vulnerability of plaques.

The aim of our study was to compare the profile of several circulating microRNAs in patients depending on CTA-based coronary artery plaque characteristics. We combined three important aspects that make it possible to assess the clinical applicability of miRNA analysis: the non-invasive nature of the study, the use of the plaque vulnerability parameters, and the association with existing methods for assessing the risk of cardiovascular complications. As a result, we have identified several candidates for the role of new biomarkers: miR-143-3p, miR-181-5p, miR-126-3p, miR-126-5p, miR-145-5p, miR-150-5p and miR-195-5p, potentially allowing risk cardiovascular restratification that might be helpful in clinical decision making.

## 2. Subjects, Materials and Methods

### 2.1. Design of the Study

In this case series study, the levels of plasma miRNAs in patients with suspected CAD, characterized by different degrees of CVR, were assessed by various criteria: prognostic scales of CVC and the presence of coronary atherosclerosis of varying severity and vulnerability based on CTA data.

The study included patients from the Clinical Center of the I. M. Sechenov First Moscow State Medical University (University Clinical Hospital No. 1), Moscow, Russian Federation. Informed voluntary consent was obtained from each participant.

### 2.2. Inclusion Criteria

I.Age range 18–80 years.II.Patients for whom multislice spiral computed tomography coronary angiography (MSCT-CA) is indicated:a.Patients with atypical anginal pain with low or intermediate CVR;b.Asymptomatic patients at high CVR;c.Patients at high CVR with indications for diagnostic coronary angiography who refused invasive tests;

III.Written informed consent of the patient to participate in the study.

### 2.3. Exclusion Criteria

I.Pregnancy and breastfeeding;II.Diabetes mellitus;III.Heart surgery or coronary interventions in anamnesis;IV.Severe heart failure (III–IV classes according to the classification of the New York Heart Association, NYHA);V.History of myocardial infarction;VI.Body mass index 35 kg/m^2^ or more;VII.The presence of any moderate or severe somatic pathology at the time of the study (including oncological diseases, impaired liver function, renal failure, systemic and inflammatory diseases, as well as any infections);VIII.Refusal of the patient from participation in the study;IX.Psychiatric disorders, including claustrophobia.

We did not include patients with diabetes mellitus in the study since they automatically belong to a very high CVR group; in addition, this allowed us to exclude the potential effect of type 2 diabetes on the levels of circulating miRNAs

### 2.4. Study Duration

The enrollment of patients was carried out from January 2020 to October 2021.

### 2.5. Description of the Intervention

Volumetric computed tomography of the coronary arteries was performed on an Aquilion ONE 640-slice computed tomography scanner (Canon, Japan) via contrast enhancement with iopromide-370 with a volume of 40–70 mL, depending on the patient’s body weight. If the patient’s heart rate was higher than 75 beats per minute, in the absence of contraindications, they were prescribed β-blockers. Analysis of computed tomography data was performed using the Vitrea software package (Canon Medical Informatics, Inc., Minnetonka, MN, USA). In the first stage, each patient underwent a quantitative assessment of coronary calcium according to the Agatson scale, then a detailed analysis of coronary arteries was carried out. In the presence of atherosclerosis-specific changes, their localization, length, structure and condition of the atherosclerotic plaque cap and artery remodeling at the site of stenosis were assessed, and the group was determined according to the coronary artery disease—reporting and data system (CAD-RADS) scale. The vulnerability of an atherosclerotic plaque was determined according to the following criteria: unevenness of the internal contour of the plaque; low plaque density (<30 Hounsfield units); presence of microcalcifications; vascular remodeling index >1.1; signs of the fibrous cap rupture [19,20]. In the absence of such changes, the plaques were considered stable. The assignment of patients with heterogeneous or mixed plaques to different groups was carried out according to the dominant morphology.

The risks of CVC in patients were assessed based on validated scales (SCORE2 and SCORE2-OP for very high CVD risk population [21,22]; Framingham [23]; ACC/AHA [24]; MESA [25]).

At the same time, with preparation for the CTA, 10 mL of venous blood was collected into tubes with ethylenediaminetetraacetic acid (EDTA) for subsequent analysis of miRNA levels in plasma. Centrifugation was carried out at room temperature with an acceleration of 1500× *g* for 15 min. Two-thirds of the plasma were taken and centrifuged for 10 min with an acceleration of 2300× *g*. About three-quarters of the plasma were taken and centrifuged again with an acceleration of 2300× *g* for 10 min. Three-quarters of the remaining plasma were poured into test tubes and frozen at −20 °C.

Frozen plasma samples were transported to the laboratory facility without thawing and then stored at −80 °C. Before miRNA isolation, plasma samples were thawed on ice and centrifuged at 16,000× g for 15 min at 4 °C to pellet down any residual cell debris. The supernatant in a volume of 300 µL was used immediately for miRNA extraction, and 10 µL was aliquoted and stored at −20 °C for further hemolysis assessment.

### 2.6. Hemolysis Assessment of Plasma Samples

Only samples without visually detected hemolysis were included in this study. Low-level red blood cell (RBC) hemolysis in plasma samples was assessed by spectrophotometric measurement of absorbance at 414 nm wavelength (peak of free hemoglobin). For each sample, 10 µL aliquot of plasma was used for hemolysis assessment within seven days after miRNA isolation. The sample was thawed, incubated at room temperature for 30 min, and analyzed on a NanoDrop^®^ 2000 spectrophotometer (Thermo Fisher Scientific, Waltham, MA, USA) by measurement of ultraviolet-visible (UV-Vis) absorbance with 1 mm path at 385 nm (A385) and 414 nm (A414) wavelengths in triplicate for each sample. For each measurement, a lipemia-independent hemolysis score (HS) was calculated based on the mean A414 and A385 values: HS = ∆(A414 − A385) + 0.16 × A385 [26]. Samples with HS > 0.25 were not included in this study. 

### 2.7. Plasma miRNA Isolation

MiRNA was isolated from 300 µL of plasma using a NucleoSpin miRNA Plasma kit (Macherey-Nagel, Düren, Germany) according to the manufacturer’s guidelines. Proteinase K digestion was performed for each plasma sample before the isolation of miRNA. Each miRNA sample had a total volume of 30 µL and was stored at −80 °C prior to cDNA synthesis.

### 2.8. cDNA Synthesis and Quantitative PCR (qPCR) for miRNA Detection

A 2 µL sample of miRNA was used for cDNA synthesis with a TaqMan Advanced miRNA cDNA Synthesis Kit (Thermo Fisher Scientific, Waltham, MA, USA), according to the manufacturer’s recommendations, using DNA Engine Tetrad 2 Thermal Cycler (Bio-Rad Laboratories, Hercules, CA, USA). Commercially available TaqMan Advanced miRNA assays with TaqMan Fast Advanced Master Mix (Thermo Fisher Scientific, Waltham, MA, USA) were used to perform qPCR, according to the manufacturer’s protocol. The list of miRNA assays with catalog numbers and mature miRNA sequences is given in Table 1. As an endogenous normalization control, hsa-miR-16-5p was chosen since it is widely used for plasma miRNA studies. For qPCR-based hemolysis assessment, a pair of miRNAs (hsa-miR-23a-3p and hsa-miR-451a) that indicate the degree of hemolysis by their ratio were included in the study. For each miRNA assay, a no-template control (NTC) containing nuclease-free water instead of a miRNA sample was analyzed. We performed qPCR using the QuantStudio 5 Real-Time PCR system (Thermo Fisher Scientific, Waltham, MA, USA) in MicroAmp 96-well PCR plates and optical adhesive film, in a “Fast” cycling mode with the following program: enzyme activation—20 s at 95 °C; 45 cycles: denature—1 s at 95 °C, anneal/extend—20 s at 60 °C. We obtained qPCR data using QuantStudio Design and Analysis Software v1.4.1 (Thermo Fisher Scientific, Waltham, MA, USA). Cq values were calculated using the automatic “Baseline” value and the experimentally set “Threshold” value of ∆Rn = 0.15 for all analyzed miRNA targets. Cq measurements were performed in a single technical replicate for each miRNA target within an individual sample. Normalization of qPCR data for each target miRNA was performed using the Cq value of hsa-miR-16-5p. For each miRNA analyzed, its plasma relative expression level was calculated as 2^−ΔCq (target miRNA—hsa-miR-16-5p)^. If the Cq value of the analyzed miRNA was undetectable by 45 qPCR cycles, its Cq was 45. The impact of RBC hemolysis on circulating miRNAs was estimated by the qPCR-detected ratio of hemolysis-dependent miRNA miR-451a abundant in RBC and miR-23a-3p, which is hemolysis-independent [27]. The difference between Cq values of these miRNAs was calculated by the formula: dCq_miR-23a-3p—miR-451a_ = Cq(hsa-miR-23a-3p) − Cq(hsa-miR-451a). For all the samples in this study, the same laboratory workflow and PCR data analysis protocol was used.

### 2.9. Statistical Data Analysis Methods

Statistical analysis was performed using Prism (Version 9.0.0 (86)) and Python 3.9.4 (pandas). The datasets were pre-processed and tested for normalcy with the Anderson–Darling test, according to which the distribution of all quantitative variables differs from the normal. Data were assessed for the presence of outliers using the ROUT method (Q = 1%).

The patient’s baseline characteristics and miRNAs plasma levels were analyzed using ANOVA for continuous variables with normal distribution, the Mann–Whitney U test for pair-wise comparison, the Kruskal–Wallis test for multiple groups for non-normally quantitative distributed variables, and the chi-square test for qualitative variables. Age- and gender-adjusted simple logistic regression was used to identify predictors of plaque development and its stability. The sensitivity and specificity of potential biomarkers were assessed via ROC analysis. Age- and gender-adjusted simple linear regression was used to identify predictors of the different risk scores values or Deming regression for different clinical and laboratory parameters. We used Holm–Sidak multiple tests correction when required. A *p*-value of less than 0.05 was considered significant for all analyses.

## 3. Results

### 3.1. Study Participants

The study included 83 patients (63% female, mean age = 65) with suspected CAD who underwent computerized tomography coronary angiography. 

Of the 83 blood plasma samples, 19 were excluded because of signs of hemolysis. In total, 64 samples were analyzed, of which two more were excluded because of too low and too high delta Cq values (miR-23a-3p-miR-451a-miRNA sensitive and insensitive to hemolysis). The final sample consisted of 62 samples (Figure 1).

The analysis of associations of miRNA levels was carried out when dividing patients into three groups according to the presence of stable and vulnerable atherosclerotic plaques and patent coronary arteries.

Patient characteristics are detailed in Table 2 and Table 3.

Patients of the study groups did not significantly differ in age and sex, calculated CVD risks determined by the SCORE2(+OP), ATP III, ACC/AHA scales, body mass index, levels of creatinine, cholesterol, high-, low- and very-low-density lipoproteins, or glomerular filtration rate. The plaque-free group did not include patients with atypical angina pectoris, but did include a significant number of patients with paroxysmal atrial fibrillation. Differences in the levels of coronary calcification (Agatson Index) were consistently observed between the groups (*p* = 0.003). In addition, in the group with vulnerable plaques, patients with higher values of glucose and triglycerides in blood plasma predominated (*p* = 0.007 and *p* = 0.03), and in the group with stable plaques, more patients received statins at the time of enrollment (*p* = 0.007). 

The levels of 15 circulating miRNAs were determined: has miR-21-5p, 23a-3p, 29b-3p, 92a-3p, 126-3p, 126-5p, 143-3p, 145-5p, 146a-5p, 150-5p, 181b -5p, 195-5p, 205-5p and 451a.

### 3.2. Association of miRNA Levels with the Type of Atherosclerotic Plaques

The relative plasma levels of various miRNAs between the groups did not differ significantly, except for the plasma level of miR 143-3p (Figure 2A) and miR 181-5p (Figure 2B). The plasma level of miR-143-3p was higher in the plaque-free group and lower in the group with vulnerable atherosclerotic plaques (*p* = 0.0046). The plasma level of miR-181b-5p was highest in the group of stable plaques compared with the group of vulnerable plaques (*p* = 0.0179).

A pair-wise comparison was performed using the Mann–Whitney U-test. Relative plasma levels (expression) data are presented as mean ± standard deviation. A *p* level < 0.05 was considered statistically significant.

ROC analysis demonstrated moderate predictive efficiency of selected microRNA (Figure 3).

### 3.3. Association of miRNA Levels with CVR

When assessing the differences in miRNA levels in patients with different risks according to the most common population scoring systems, no significant differences were obtained for SCORE2 (+OP) and ATP III. When evaluated by ACC/AHA, significant (*p* < 0.05) differences in miR-126-5p and 150-5p levels were obtained for patients at risk > 10% (*n* = 41) versus risk < 10% (*n* = 21) (Table 4).

The levels of miR-195-5p were significantly higher in patients with a MESA risk > 7.5% (*p* < 0.05). The levels of this miRNA were also higher in patients with a calcium index = 0 (*p* = 0.039). Levels of circulating miR-181b-5p were higher (*p* = 0.036) in patients with high levels of coronary calcification (>100 Agatson units).

To assess the possible linear relationship between the relative plasma levels of the selected miRNAs and the values of the final risk factors on various scales, Deming’s regression was used, considering the correction for multiple hypothesis testing. We found no statistically significant data for the SCORE2(+OP) and ATP III scales. Statistically significant linear relationships were demonstrated with CVR measured by the ACC/AHA scale for miR 145-5p (*p*-value = 0.0011, adjusted *p*-value = 0.0164), and with the level of coronary calcium for miR 126-3p (*p*-value = 0.0033, adjusted *p*-value = 0.0484) (Figure 4 and Figure 5).

## 4. Discussion

According to our results, two miRNAs can be associated with the stability (or vulnerability) of the atherosclerotic plaque according to CTA data. These are miR-143-3p, which showed the lowest relative plasma levels in the vulnerable plaque group and miR-181-5p, which showed the highest plasma level in the stable plaque group.

The biological role of miRNA in the processes of athero- and antiatherogenesis is being actively studied. Thus, miR-143-3p is involved in the regulation of endothelial and vascular smooth muscle cells (VSMC) functions [28]. Analysis performed by Degano et al. in the REGICOR study describes the role of miR-143-3p in the inflammatory response to acute myocardial injury, as indicated by the data on the increase in the concentration of this miRNA in the blood of patients immediately after a vascular event [29]. The dual function of miR-143-3p is noted in the processes of stabilization and destabilization of atherosclerotic lesions. The analysis of the target genetic mechanisms of regulation of vascular inflammation, carried out by the authors, shows a rather wide spectrum of miR-143-3p activity. Thus, MAPK7, PLPP3, SERPINE1 and THBS1, which are involved in the activation of endothelial cells, CTG and NRG1, which are involved in their proliferation, migration, and apoptosis, and SENP2, which regulates the response to blood flow characteristics, are described as target genes. In addition, miR-143-3p affects the expression of CACNA1C, COL1A1, COL5A2 and PTGS2 genes of VSMC, which are responsible for the processes of proliferation and synthesis of the extracellular matrix. [29,30]. In our study, an increase in miR-143-3p plasma level in the plaque-free group is noted (Figure 2A), which is consistent with the data of Degano et al. and may indicate the protective role of this miRNA on patent arteries. At the same time, within the framework of the study of miR-143-3p and coronary atherosclerosis, no differentiation was made between vulnerable and stable atherosclerotic plaques, and with the development of myocardial infarction, the levels of miR-143-3p in blood plasma increase again [29]. Additionally, our data are consistent with the results obtained by Gonzalo-Calvo et al.; in their study, the levels of circulating miR-143-3p inversely correlated with the severity of coronary atherosclerosis but did not show significance as biomarkers. However, the authors did not analyze the direct characteristics of atherosclerotic plaques, which could affect the result of the study [31]. Analyzing the data obtained by us and in recent studies described above, it can be assumed that the participation of miR-143-3p in the processes of atherosclerosis and destabilization of an atherosclerotic plaque differs at different stages, and the deviation of the levels of this miRNA can be used as a marker of plaque instability and a harbinger of impending CVC.

Another miRNA, which in our study (Figure 2B) showed a relationship with the type of atherosclerotic plaque–miR-181b-5p–might be considered atheroprotective [32]. The main targets of this miRNA, importin-α3, PI3K, MAPK and KPNA4, are involved in vascular inflammation reduction and stimulating the proliferation and migration of VSMC [33,34]. Thus, in vivo studies showed that overexpression of miR-181b-5p led to a decrease in the atherosclerotic plaque vulnerability through the regulation of Notch1 and the transcription factor NF-kB [35]. However, in the study of carotid plaques, the expression level was higher in the group without atherosclerosis in one study [36] and, conversely, in vulnerable plaques in another [37]. Such differences can be explained by studies of different vascular territories, estimations of miRNA levels in tissues and in macrophages and exclusively molecular criteria for atherosclerotic plaque vulnerability. Most likely, the dynamics of these miRNA levels depend on the degree of coronary calcification, with which a strong correlation is also noted. The phenomenon of the relationship between miR-181b-5p and vascular calcification may be due to the already known mechanisms of atheroprotection but requires further study. It is possible that deviations in the levels of this miRNA, along with miR-141-3p, can be used as a marker of the stability of atherosclerotic lesions. An atheroprotective role of miR-181b-5p is also suggested in a Patterson et al.’s study, where they analyzed a large pool of miRNAs expression using the pathway-specific microarray profiling [38]. The expression of this miRNA was higher in patients with coronary atherosclerosis and a low risk of CVC, according to the Framingham score. In the same group of patients, there was an increase in miR-21-5p expression, which in our study was associated with a high risk of complications according to ACC/AHA. Interestingly, that many miRNAs from Patterson’s et. al. study and in our work (miR-23a-3p, -92a-3p, -126-3p, -143-3p, -145-5p, -146a-5p, -150-5p and -195-5p), did not reveal any associations with CVR or atherosclerosis. Such results may be due to different methods of miRNA analysis, the absence of other scales for assessing vascular risks, coronary calcium, and the identification of a vulnerable plaques group. In another study of tissue-specific miRNAs in patients with stable and unstable CAD, there were significant differences in the expression of miR-29b-3p, -92a-3p, -126-3p, -126-5p, -143-3p, -145-5p, -146a-5p, -150-5p, -195-5p and a number of others; however, none of the overlapping with our work miRNAs was validated at subsequent stages [17]. The fundamental difference of our work is the assessment of circulating miRNAs rather than tissue-specific ones since they are most applicable as biomarkers. Patterson et al. showed that miR-451a expression was downregulated in high CVR patients without CAD [38], and Li S. et al. supposed that this miRNA can be a biomarker of early plaque rupture [18]. However, these studies did not take into account the hemolysis level, which can significantly affect the levels of circulating miRNAs, and since the level of miR-451a, as mentioned above, is hemolysis-dependent, the associations obtained with it are uncertain.

In a study by Zampetaki et al., there was indicated the potential (diagnostic significance was not achieved) ability of miR-126, -197 and -223 to increase the accuracy of CVR assessment according to the Framingham scale [13]. The study was carried out in 2012 when miR-126-3p and miR-126-5p were not considered separately in all studies. However, these data are consistent with our findings on the ACC/AHA risk association with miR-126-5p.

MiRNAs 126-5p and -3p are within the same miRNA family but differ in their functions. Thus, when studying the effect of miRNA on the proliferation of VSMCs in vivo, miR-126-3p had a negative effect on this process, stimulating atherogenesis, while miR-126-5p had an atheroprotective effect. [39]. The protective mechanism of miR-126-5p is presumably realized through suppression of the DLK1 protein, which inhibits a number of signaling pathways that regulate the proliferation of VSMCs [40]. Interestingly, the mechanism of antiatherogenic action of miR-126-5p described by Santovito et al. is based on direct inhibition of the caspase-3 gene in the cell nucleus without participation in the RISK complex, which leads to inhibition of the processes of apoptosis of endothelial cells and atherosclerosis [41]. Another potential mechanism of miR-126-5p action is inhibition of the MAPK/ERK pathway, which is one of the pathogenetic factors of vascular inflammation [42]. In addition, changes in shear stress in atherosclerosis injured arteries lead to increased expression of miR-126-5p without affecting miR-126-3p [43,44]. In turn, shear stress activates NO synthase, leading to a local improvement in blood flow and counteracting atherosclerosis [45].

As for miR-126-3p, our study showed a statistically significant linear association of its levels with the level of coronary calcium (Figure 5). The mechanism of participation of this miRNA in the process of arterial calcification is poorly understood, but it is believed that it is based on the above-mentioned MAPK/ERK pathway regulation [46]. The results of our study are consistent with the data of Chen et al., where along with a number of pro-inflammatory markers and miR-146a, a significant increase in the expression of miR126-3p in patients with coronary atherosclerosis and a positive correlation with the level of coronary calcium were found [47]. In our study, miR-146-5p did not show associations with coronary calcium, and the correlation of this miRNA with a risk of >10% or >7.5% by ACC/AHA is noted at the border of statistical significance (Table 4).

Another miRNA that has shown a strong association with CVR > 10% by ACC/AHA is miR150-5p (Table 4). Studies of the association of these miRNA levels with the calculated CVR have not been carried out; however, this miRNA has shown significance as a biomarker of unstable angina pectoris and acute coronary syndrome, both as combined markers [48,49] and single [50]. 

The pathophysiological role of miR-150-5p is controversial. Thus, a number of in vitro and in vivo studies show that it helps to reduce the levels of pro-inflammatory cytokines IL-1b, IL-6 and other factors [51] and also inhibits the proatherosclerotic transcription factor c-Myb affecting chemotaxis and migration of monocytes [52], which leads to protection of the myocardium from injury [53], overload and subsequent fibrosis [52,54,55]. However, other studies point to miR-150-5p activity leading to vascular inflammation, endothelial cell apoptosis and progression of atherosclerosis [56,57]. Rayner et al., in the analysis of the features of the activity of pro-inflammatory miRNAs, indicate the possible pro-inflammatory effect of miR-150 secreted by monocytes in special exosomes [58]. Another study, carried out on isolated human cells, expands the understanding of the potential activity of miR-150, indicating not only its involvement in the process of inflammation but also in the process of lipid concentration and metabolism [59].

Our data on the levels of circulating miR-195-5p and 145-5p in patients from groups of different calculated CVR were described for the first time.

In our pilot study, decreased miR-195-5p levels were associated with a higher CVR by the MESA scale [60]. However, in a small sample, comparisons were made between borderline and moderate risk groups, in which there was no significant difference in the level of coronary calcification. Upon validation, we obtained an increase in miR-195-5p level in patients with moderate and high CVR. This is comparable with the data of other studies on this miRNA, the levels of which in the blood plasma of patients with myocardial infarction were higher compared to the control group [25,48]. 

The biological role of miR-195-5p also appears to be controversial. A number of studies indicate its anti-inflammatory activity through a decrease in the pathological proliferation, migration of VSMCs and the formation of neointima [61]. In addition, miR-195-5p inhibits the pro-inflammatory activity of macrophages, which ultimately provides the antiatherogenic function of miR-195 [62]. In studies of heart failure [63] and myocardial hypertrophy [64], miR-195 levels increase with disease progression, while there is no common understanding of the direct role of this miRNA. Thus, the researchers Okada et al. and Ding et al., in their studies on biological models, draw diametrically opposite conclusions about the need to inhibit or stimulate the expression of miR-195-5p for heart failure treatment [65,66], and Cheng et al. propose to use miR-195-5p as a treatment for ischemic and hemorrhagic strokes [67].

The role of miR-145-5p in the pathophysiology of atherosclerosis, atherosclerotic plaque rupture and the development of CVC has been studied relatively extensively. Most studies point to a protective function as it leads to a decrease in the proliferation of VSMCs by inhibition of the CD40 receptor and a decrease in IL-6 levels [68,69]. Other ways of VSMC activity regulation described the effect of miR-145 on calcium calmodulin-dependent protein kinase II (CaMKII) contained in cardiomyocytes [70], SMAD4 proteins [71], KLF4, KLF5 and ACE receptors [72]. The latter can explain the association of this miRNA with CVR by ACC/AHA (Figure 4) since on the American scale, hypertension treatment is considered one of the variables. 

Decreased expression of miR-145-5p led to the progression and rupture of atherosclerotic plaques in vivo and in vitro [73]. At the same time, the injection of miR-145-5p in an in vivo model markedly reduced the size of plaques in the aorta and brachiocephalic arteries. The area of the fibrous capsule increased significantly, the volume of the necrotic cores decreased and the collagen content in plaques increased. [74]. Additionally, when miR-145 is overexpressed, fewer myocardial damage is noted [70], and there is a lesser susceptibility of the injured myocardium to the development of ventricular tachycardia [71]. Controversially, an increase in miRNA expression is observed in patients with arterial hypertension [75], atherosclerosis [76] and myocardial infarction [77].

Summarizing the data obtained, we can suppose that the cardioprotective role of miR-145-5p, and its increase as atherosclerosis progresses, is most likely a compensatory mechanism. It can be assumed that the decrease in the levels of miR-145-5p in patients with atherosclerosis will be a predictor of a poor prognosis.

### Limitations

The main limitations of the study are the small proportion of truly vulnerable plaques without the formation of areas of more pronounced calcification (it would be ideal for comparing plaques with zero coronary calcification), as well as the small sample size. In the plaque-free group, there was a significantly higher proportion of patients with atrial fibrillation, which could indirectly affect miRNA plasma levels. In addition, the study noted a high percentage (22.9%) of exclusion of samples because of hemolysis, which can be explained both by the thoroughness of the selection criteria and the peculiarities of simultaneous blood sampling. Moderate sensitivity and specificity of miR-143-3p and miR-181b-5p related plasma levels are due to sampling sizes and study design. Additionally, our study was not prospective; therefore, we cannot assess the impact of circulating miRNAs on CVR.

## 5. Conclusions

Our data suggest the role of miR-143-3p and miR-181b-5p as biological markers of the course of the atherosclerotic process. Several other miRNAs such as miR-126-3p, 126-5p, 145-5p, 150-5p, and 195-5p can be used as potential modifiers of the estimated risk of CVC.

However, to apply the analysis of circulating miRNA levels in clinical practice, it is necessary to validate the results obtained in large prospective clinical trials.

## Figures and Tables

**Figure 1 ncrna-08-00047-f001:**
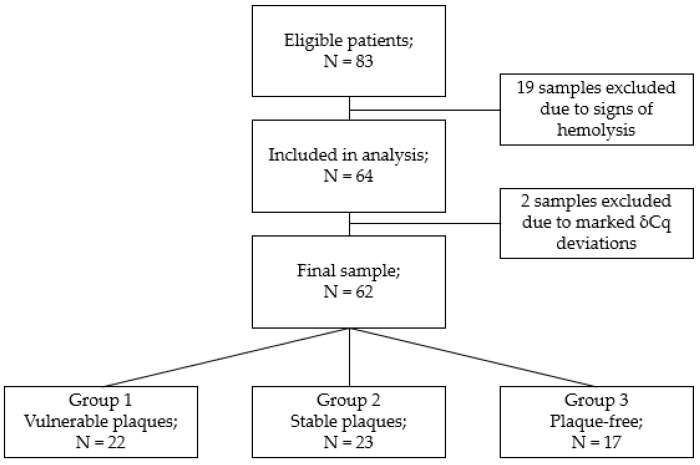
Block diagram of the study.

**Figure 2 ncrna-08-00047-f002:**
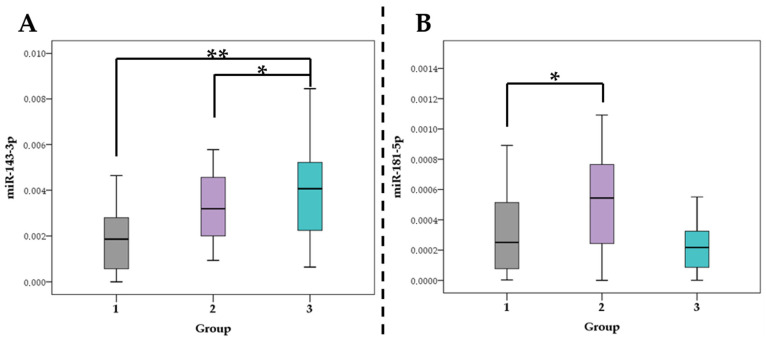
Results of comparison of miRNA relative plasma level in patients with different characteristics of atherosclerotic plaques (nonparametric test for comparison of multiple Kruskal–Wallis groups due to data distribution different from normal). (**A**) miR-143-3p relative levels. * *p*-value = 0.0457, ** *p*-value = 0.0046; (**B**) miR-181b-5p relative levels. * *p*-value = 0.0179.

**Figure 3 ncrna-08-00047-f003:**
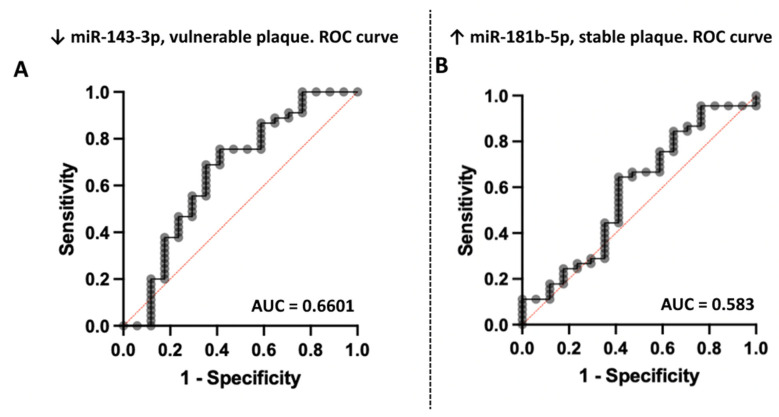
Results of sensitivity and specificity analysis of (**A**) miR-143-3p relative plasma levels decrease as a marker of plaque vulnerability and (**B**) increase in miR-181b-5p levels as a marker of plaque stability. ROC curves.

**Figure 4 ncrna-08-00047-f004:**
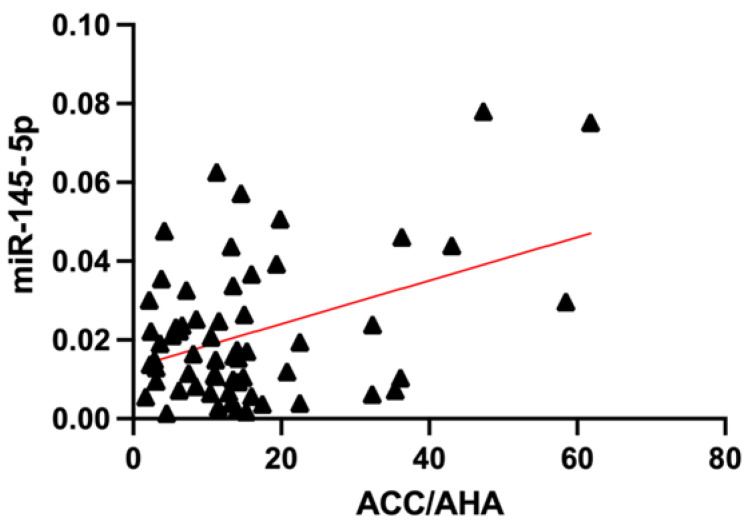
Relationship of miR-145-5p relative plasma levels in patients with different values of CVR estimated by ACC/AHA.

**Figure 5 ncrna-08-00047-f005:**
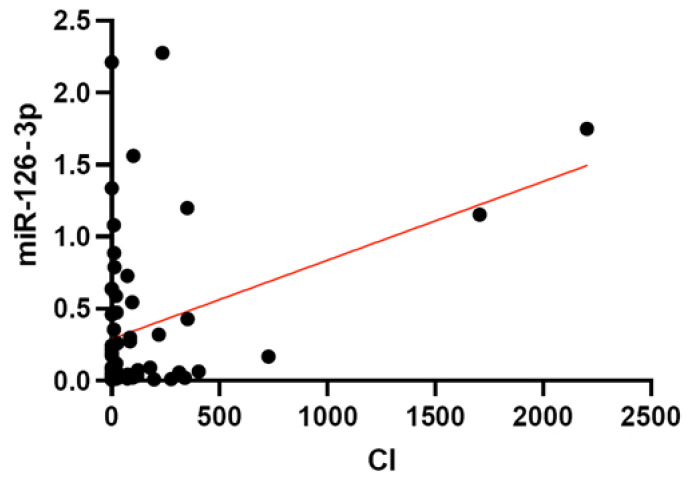
Relationship of miR-126-3p relative plasma levels in patients with different values of coronary calcification according to the Agatson scale (CI).

**Table 1 ncrna-08-00047-t001:** List of the miRNA assays used for qPCR.

Assay Name	Assay ID	Mature miRNA Sequence	Type of miRNA
hsa-miR-16-5p	477860_mir	UAGCAGCACGUAAAUAUUGGCG	Normalization control
hsa-miR-23a-3p	478532_mir	AUCACAUUGCCAGGGAUUUCC	Hemolysis assessment
hsa-miR-451a	478107_mir	AAACCGUUACCAUUACUGAGUU	Hemolysis assessment
hsa-miR-126-3p	477887_mir	UCGUACCGUGAGUAAUAAUGCG	Candidate to atherosclerosis
hsa-miR-126-5p	477888_mir	CAUUAUUACUUUUGGUACGCG	Candidate to atherosclerosis
hsa-miR-143-3p	477912_mir	UGAGAUGAAGCACUGUAGCUC	Candidate to atherosclerosis
hsa-miR-145-5p	477916_mir	GUCCAGUUUUCCCAGGAAUCCCU	Candidate to atherosclerosis
hsa-miR-146a-5p	478399_mir	UGAGAACUGAAUUCCAUGGGUU	Candidate to atherosclerosis
hsa-miR-150-5p	477918_mir	UCUCCCAACCCUUGUACCAGUG	Candidate to atherosclerosis
hsa-miR-181b-5p	478583_mir	AACAUUCAUUGCUGUCGGUGGGU	Candidate to atherosclerosis
hsa-miR-195-5p	477957_mir	UAGCAGCACAGAAAUAUUGGC	Candidate to atherosclerosis
hsa-miR-205-5p	477967_mir	UCCUUCAUUCCACCGGAGUCUG	Candidate to atherosclerosis
hsa-miR-21-5p	477975_mir	UAGCUUAUCAGACUGAUGUUGA	Candidate to atherosclerosis
hsa-miR-223-3p	477983_mir	UGUCAGUUUGUCAAAUACCCCA	Candidate to atherosclerosis
hsa-miR-29b-3p	478369_mir	UAGCACCAUUUGAAAUCAGUGUU	Candidate to atherosclerosis
hsa-miR-92a-3p	477827_mir	UAUUGCACUUGUCCCGGCCUGU	Candidate to atherosclerosis

**Table 2 ncrna-08-00047-t002:** Parametric characteristics of patients.

Parameter ^a^	Group 1 ^b^	Group 2 ^c^	Group 3 ^d^	SD	ANOVA
Age (years)	62.18	67.09	65.94	9.70	0.217
BMI (kg/m^2^)	30.04	27.69	28.36	4.41	0.152
ATP III	9.77	10.16	8.14	6.64	0.622
ACC/AHA	12.02	17.04	17.38	13.06	0.333
MESA	9.26	14.52	3.62	7.13	<0.001
Agatson index	45.50	336.78	0.00	362.83	0.003
Glucose (mmol/L)	5.81	5.16	5.18	0.78	0.007
Creatinine (µmol/L)	88.42	87.73	86.46	15.14	0.925
eGFR_CKD-EPI_ (mL/min/1.73 m^2^)	69.43	64.90	67.14	15.06	0.609
Cholesterol (mmol/L)	5.94	5.34	5.73	1.28	0.287
Triglycerides (mmol/L)	1.92	1.38	1.39	0.77	0.030
LDL (mmol/L)	3.78	3.33	3.61	1.16	0.424
HDL (mmol/L)	1.33	1.42	1.56	0.38	0.162
VLDL (mmol/L)	0.84	0.58	0.66	0.39	0.192

^a^ BMI—body mass index; eGFR—estimated glomerular filtration rate; LDL—low-density lipoproteins; HDL—high-density lipoproteins; VLDL—very-low-density lipoproteins. ^b^ Group 1—patients with predominant vulnerable atherosclerotic plaques; ^c^ Group 2—patients with predominant stable atherosclerotic plaques; ^d^ Group 3—patients with patent coronary arteries (plaque-free group).

**Table 3 ncrna-08-00047-t003:** Nonparametric characteristics of patients.

Parameter, % ^a^	Group 1 ^b^	Group 2 ^c^	Group 3 ^d^	χ2	P
Female sex	59.09	82.61	76.47	3.30	0.192
Angina atypical	45.45	34.78	0.00	10.20	0.006
Hypertension	95.45	82.61	88.24	1.86	0.395
Hypertension 3 grade	50.00	43.48	41.18	0.34	0.842
Positive stress-test	40.91	60.87	23.53	5.64	0.060
EF > 50%	90.91	95.65	88.24	1.37	0.679
Paroxysmal AFib	9.09	13.04	41.18	3.29	0.027
Statins	36.36	73.91	41.18	8.23	0.025
Smoking	18.18	13.04	5.88	2.19	0.524
ACE inhibitors	31.82	34.78	23.53	5.21	0.739
ARBs	36.36	52.17	29.41	6.85	0.312
Beta blockers	72.73	52.17	47.06	7.13	0.211
Calcium channels blockers	36.36	39.13	11.76	5.21	0.137
Oral anticoagulants	13.64	4.35	35.29	2.74	0.029
Acetylsalicylic acid	31.82	47.83	35.29	6.58	0.514

^a^; EF—ejection fraction; AFib—atrial fibrillation; ACE—angiotensin-converting enzyme; ARBs—angiotensin receptor blockers. ^b^ Group 1—patients with predominant vulnerable atherosclerotic plaques; ^c^ Group 2—patients with predominant stable atherosclerotic plaques; ^d^ Group 3—patients with patent coronary arteries (plaque-free group).

**Table 4 ncrna-08-00047-t004:** Association of miRNA levels with the cardiovascular risks estimated by ACC/AHA.

	ACC/AHA >10%		ACC/AHA >7.5%	
	*p*-value	Adjusted *p*-value	*p*-value	Adjusted *p*-value
miRNA 126-5p	0.003	0.0412	0.110	0.4773
miRNA 21-5p	0.014	0.1453	0.024	0.3054
miRNA 146a-5p	0.019	0.1746	0.041	0.3949
miRNA 92a-3p	0.013	0.1453	0.094	0.4773
miRNA 150-5p	<0.001	0.0149	0.025	0.3054
miRNA 181b-5p	0.023	0.1889	0.121	0.4773

## Data Availability

Not applicable.

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
