# Peer review of "The Profile of Circulating Blood microRNAs in Outpatients with Vulnerable and Stable Atherosclerotic Plaques: Associations with Cardiovascular Risks"

_ncrna, 2022, doi:10.3390/ncrna8040047_

Round 1

Reviewer 1 Report

See attached!!

Author Response

I am writing to submit our revision of the manuscript entitled “The profile of circulating blood microRNAs in outpatients with vulnerable and stable atherosclerotic plaques. Associations with cardiovascular risks” for consideration for publication in the Journal

We thank you for reviewing our manuscript and for your comments.

When editing our manuscript, we took into account the outlined comments.

The answers to the several comments are in the attachment

Reviewer 2 Report

Dear Editor, 

Thank you very much for giving me the chance to review this paper.

The manuscript by Rozhkov et al have examined the levels of certain microRNAs in cases with cardiovascular disease. The study was conducted in with group with vulnerable atherosclerotic plaques, group with stable atherosclerotic plaques and a group without atherosclerotic plaque. They used the coronary computed tomography angiography to diagnose the atherosclerotic plaques. Then they have evaluated the associations of miRNA levels with risks to cardiovascular based on scales SCORE2 (+ OP), ACC/AHA, ATP-III and MESA. The plasma levels of microRNA were estimated with real time polymerase chain reaction. The results indicated that there was significant differences in miR-143-3p levels in plaque-free vs vulnerable plaque groups) and miR-181b-5p in stable vs. vulnerable plaques groups). They proposed that that miR-143-3p and miR-181-5p are potential biomarkers of atherosclerosis, and that that miR-143-3p and miR-181-5p may be regarded as potential biomarkers. The paper is fine however; I suggest that the following points to be considered before acceptance of the manuscript. 

1- The Materials and Methods should be replaced to Subjects, Materials and Methods. 

2- The inclusion and exclusion criteria should be rephrased.

3- What is the difference between Non-inclusion criteria and exclusion criteria?

4- The 3.4. Adverse events in line 285 should be removed. 

5- The limitations of the study should be included in last part of the discussion. 

6- The figures and tables should also be cited in the discussion. 

7- The authors should re write the discussion and cite the references immediately after the statement instead not making 4 or 5 line and then 1 citation.

8- the last reference in the text is no 56, while in the list 70! How come?

9- What is the unit in the Y axis in figure 2, 3 and 4.

10- the manuscript should be re written with careful citation of figures and references.   

Author Response

(The authors gave the same response as above.)

Round 2

Reviewer 1 Report

Thank you so much for providing the revised version of your paper.  I took note of the revised paper that the authors tried to address my concerns.